# Monitoring Soil Surface Mineralogy at Different Moisture Conditions Using Visible Near-Infrared Spectroscopy Data

**Irena Ymeti \*, Dhruba Pikha Shrestha and Freek van der Meer**

Department of Earth Systems Analysis, Faculty of Geo-Information Science and Earth Observation, University of Twente, P.O. Box 217, 7500 AE Enschede, The Netherlands; d.b.p.shrestha@utwente.nl (D.P.S.); f.d.vandermeer@utwente.nl (F.v.d.M.)

**\*** Correspondence: i.ymeti@utwente.nl

**Abstract:** The Soil minerals determine essential Soil properties such as the cation exchange capacity, texture, structure, and their capacity to form bonds with organic matter. Any alteration of these organo-mineral interactions due to the Soil moisture variations needs attention. Visible near-infrared imaging spectroscopy is capable of assessing spectral Soil constituents that are responsible for the organo-mineral interactions. In this study, we hypothesized that the alterations of the surface Soil mineralogy occur due to the moisture variations. For eight weeks, under laboratory conditions, imaging spectroscopy data were collected on a 72 h basis for three Silty Loam soils varying in the organic matter (no, low and high) placed at the drying-field capacity, field capacity and waterlogging-field capacity treatments. Using the Spectral Information Divergence image classifier, the image area occupied by the Mg-clinochlore, goethite, quartz coated 50% by goethite, hematite dimorphous with maghemite was detected and quantified (percentage). Our results showed these minerals behaved differently, depending on the Soil type and Soil treatment. While for the soils with organic matter, the mineralogical alterations were evident at the field capacity state, for the one with no organic matter, these changes were insignificant. Using imaging spectroscopy data on the Silty Loam soil, we showed that the surface mineralogy changes over time due to the moisture conditions.

**Keywords:** Soil mineralogy; imaging spectroscopy; visible near-infrared; Spectral Information Divergence; drying-field capacity; field capacity; waterlogging-field capacity

## 1. Introduction

The Soil minerals vary extensively in their composition, crystallinity and charge characteristics. Therefore, they determine essential Soil properties such as the cation exchange capacity (CEC), texture, structure, and their capacity to form bonds with organic matter (OM). The mobilization and precipitation of the cations such as $Si^{4+}$, $Fe^{3+}$, $Al^{3+}$, $Mg^{2+}$, and $Ca^{2+}$ can promote the disaggregation and breakdown of the Soil aggregates as well as the formation of new ones. The soils containing polyvalent cations such as $Ca^{2+}$, $Al^{3+}$, and $Fe^{3+}$ are resistant to slaking [1,2]. Indeed, the interactions between the mineral particles and the OM in Soil depend on the concentration of these cations [3]. Iron hydroxides such as the goethite, hematite, or maghemite can interact with both the clay minerals and organic compounds to form clay–mineral–organic associations, acting as binding and cementing agents in the soil [4,5]. Moreover, the amorphous iron oxides, are more effective than the crystalline Fe oxides in stabilizing the soil, due to their large and reactive surface area [6].

The other divalent cation, $Mg^{2+}$, has an excellent hydration radius, which enables the Soil to absorb more water [7]. As a result, the van der Waals forces that hold the Soil particles together weaken, affecting the soil structure.

Recently, in their review, [8] pointed out that few studies have considered the effect of the mineral composition on the formation and turnover of micro-aggregates. Using a linear regression model for analyzing the biotic and abiotic contributions to Soil stability, [9] found that the abiotic factors played a more critical role in determining the Soil stability than did biotic factors. Moreover, recent studies have shown that the iron hydroxides and clay particles are the dominant mechanisms for the Soil formation at the microscale (<250 μm) [10,11]. The interactions of the organo-mineral hydroxide, organo-phyllosilicate clay mineral and phyllosilicate-hydroxide mineral occurring in the Soil vary with mineral structure, Soil solution ionic strength and pH [12]. Any alteration of these organo-mineral interactions due to the moisture variations needs to be identified [13].

The imaging spectroscopy operating in visible, near-infrared and shortwave infrared (VNIR-SWIR) regions of the electromagnetic spectrum can assess several Soil properties such as the OM, iron, clay, calcium carbonate [14–16]. To quantify the mineralogical abundances of complex Soil mixtures at wavelength region, 2100–2400 nm, [17] combined the absorption features parametrized by exponential Gaussian optimization with the regression tree approach. They have shown that the SWIR spectra (2100–2400 nm) may be used for quantifying the mineral abundance of complex Soil mixtures. Another study used the proximal remote sensing in the VNIR-SWIR and mid-infrared (MIR) region to estimate the weathering indices [18]. These indices express the molar ratios of the immobile to mobile mineral oxides present in the soil. Using the partial least squared regression (PLSR) approach for spectral calibration and prediction, they reported that the best weathering index predictions were derived when the VNIR-SWIR and MIR were combined, and significant spectral features selected for analysis. Also, knowing that many spectrally active Soil constituents are responsible for the Soil aggregation, [19] combined the PLSR approach with Diffuse Reflectance Spectroscopy to estimate the Soil aggregate characteristics such as the Geometric Mean Diameter which is a quantitative descriptor of the Soil structure. Their results showed that the geometric mean diameter and the median aggregate size parameters provided good predictions with the ratio of performance deviation (RPD) values ranging from 1.99 to 2.28.

The VNIR imaging spectroscopy is capable of assessing the spectral Soil constituents, that are responsible for the organo-mineral interactions. These interaction mechanisms occurring naturally in the Soil depend not only on the Soil mineralogy and their reactive surface area, Soil type and Soil texture but also on the moisture condition. However, there is limited knowledge of the Soil mineralogical behavior at different moisture content occurring in a short period. Therefore, in this study, we hypothesized that the alterations of the Soil surface mineralogy occur due to the moisture variations. To test our hypothesis, we scanned the Silty Loam Soil samples varying in the organic matter (no, low, and high), under drying-field capacity, field capacity and waterlogging state in laboratory conditions at a micro-plot scale at 72 h basis using an imaging spectrometer camera for eight weeks.

## 2. Materials and Methods

### 2.1. Experimental Setup

The Silty Loam soils with various OM contents and added hematite were exposed to different Soil moisture conditions to study their mineralogical changes over time. We designed a laboratory experiment consisting of an imaging spectrometer camera, a light source and a sliding table integrated into a fixed setup. Also, on the sliding table, a tray filled with Soil was placed. Two Soil samples, the Silty Loam with low and high OM content (Soil 2 and Soil 3), were collected from the topSoil (20 cm) of two different agricultural fields in Limburg, The Netherlands. The agricultural crop cultivated in both fields was maize. However, at the time of Soil sampling, this crop was already harvested. These Soil samples were collected at the end of September 2014. We chose the Silty Loam soils because they have low aggregate stability [20]. The other Soil sample was obtained by destroying the OM of Soil 2 by heating it at 550 °C for at least 12 h. The new Soil sample with no OM was mixed manually with hematite 0.5% concentration obtaining Soil 1. Besides the loss of OM at 550 °C, the clay mineral

kaolinite decomposes above this temperature leading to the amorphous alumino-silicate material [21]. Likewise, the Soil mineral goethite transforms to hematite or maghemite at the temperature ranging from 250 °C to 420 °C [22,23]. However, these mineralogical alterations of Soil 1 were out the scope of this study. Each of these Soil samples was triplicated. Table 1 summarized the Soil characteristics used in this study.

**Table 1.** The soils used in this study. Soils 2-3 (low and high organic matter (OM)) were collected from Limburg province in The Netherlands. Soil 1 (no OM and added hematite) was obtained from Soil 2.

| Soil ID | Soil Particle Size (%) | | | Texture Class | OM (%) | $Fe_2O_3$ (%) |
|---------|------|------|------|---------------|--------|---------------|
| | **Clay** | **Silt** | **Sand** | | | |
| Soil 1 | 16 | 71 | 13 [1] | Silty Loam | 0 | 0.5 |
| Soil 2 | 16 | 71 | 13 | Silty Loam | 4.6 [2] | Na [4] |
| Soil 3 | 23 | 52 | 25 | Silty Loam | 12.3 [3] | Na |

[1] The determination of the Soil particle size was only performed before the Soil sample was placed at 550 °C. [2] The OM was determined by heating the sample at 550 °C for more than 12 h and calculating the weight loss on the dry soil. [3] The high OM content in Soil 3 is probably coming from sewage sludge manure mixed with plant residues application by the farmer some days before Soil sample was taken. [4] Not applicable.

The Soil samples remained at room conditions for four weeks. Since our goal was to investigate the Soil mineralogical weathering at different Soil moisture conditions, twenty-seven plastic trays of $15 \times 9 \times 1$ cm$^3$ were filled manually with Soil using a small shovel. As a result, the Soil aggregates of various sizes occurred in the tray. In this study, three Soil treatments, the drying-field capacity (D-FC), field capacity (FC) and waterlogging-field capacity (W-FC), were considered as separate setups. Therefore, each Soil treatment had its own set of Soil samples triplicated (Soil 1–Soil 3). The air-dried soils were weighted using a balance. To place them at the field capacity, we added deionized water corresponding to 57%, 54%, and 75% of the weight of Soil 1, Soil 2, and Soil 3, respectively. The deionized water was added carefully at the edge of the trays.

At the D-FC Soil treatment, nine Soil samples experienced repeated cycles of drying and field capacity. First, the Soil samples were placed in the oven at 40 °C for 72 h. Next, they were taken out to set at the FC using 200 mL and 240 mL of deionized water for Soil 1–2 and Soil 3, respectively. Afterward, the Soil samples were lifted to allow the excess water to leak out. In this way, we obtained the Soil samples at the FC. After staying at the field capacity for 72 h, the Soil samples were placed again in the oven at 40 °C. We repeated this exchange between drying and field capacity state until the end of the experiment.

To keep nine Soil samples at the FC, we added 20 mL of deionized water. Later, the Soil samples were carefully lifted to enable the leaking of the excess water. They were considered at the FC when the leaking process had stopped. We repeated this procedure every 72 h for the entire period that the experiment ran.

At the W-FC treatment, nine Soil samples went to repeated cycles of waterlogging and field capacity state. First, the Soil samples were placed at the waterlogging by saturating them with 370 mL, 310 mL and 410 mL of deionized water for Soil 1, Soil 2 and Soil 3, respectively. They stayed in this condition for 72 h. After 72 h, the Soil samples were carefully lifted to allow the water to leak out until the FC was reached. Then, the Soil samples remained at the FC for 72 h. Next, they were placed again at the waterlogging state. We repeated this cycle for the entire period that the experiment ran.

The drainage was enabled by 5 mm diameters holes drilled at the bottom of the tray. Likewise, the pantyhose filters were used to avoid the Soil leaking out. The excess water was collected to determine the loss of soluble cations of each Soil sample (see Section 2.5).

Moreover, to eliminate any external influence on the Soil samples, they were covered with a plastic lid. At the D-FC and FC treatments, the Soil samples were scanned every 72 h. At the D-FC, the Soil samples were scanned either at the drying or field capacity state. Moreover, at the FC treatment, the Soil samples were always scanned at the field capacity. Likewise, the Soil samples were scanned at

the field capacity every 144 h at the W-FC treatment. The images at the waterlogging conditions were discarded for further analysis because the water causes dispersion, making the data not comparable to the others with no standing water. Twenty-seven Soil samples filled with Soil with different OM content and the one with added hematite were scanned by the VNIR imaging spectrometer camera, as shown in Figure 1. We collected the images for eight weeks. The decision of eight weeks experiment was related to the laboratory facilities.

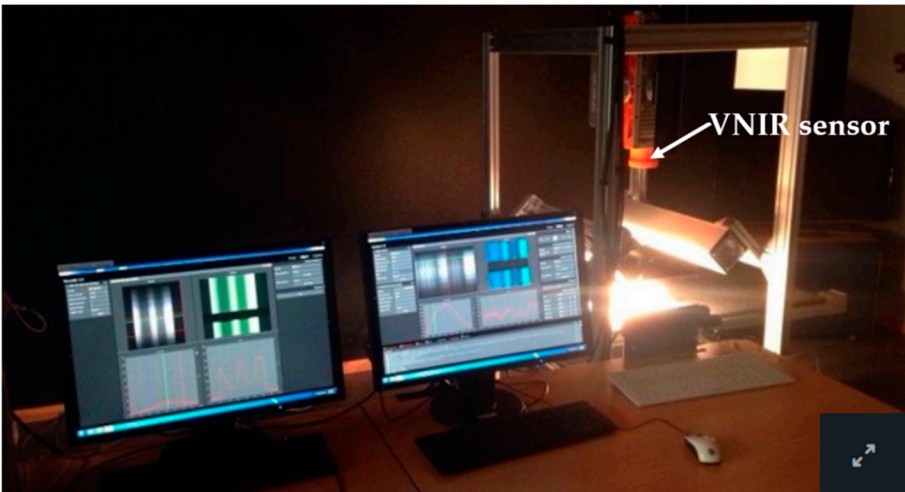

**Figure 1.** The experimental laboratory setup for image data collection. On the tripod in the center is the VNIR imaging spectrometer camera placed at an angle of 90°. Next to the sensor is a sliding table where the Soil tray is placed for scanning. On the right and the left side of the sliding table, an external light source is integrated to illuminate the tray during image acquisition.

### 2.2. Image Acquisition

To investigate the mineralogical changes on the Soil surface under controlled laboratory conditions, we used a Specim imaging spectrometer camera. This camera works as a push-broom scanner and provides contiguous spectral information for each pixel [24]. Table 2 summarizes the Specim camera characteristics.

**Table 2.** Specim imaging spectrometer camera characteristics of the visible near-infrared (VNIR) sensor.

| Characteristics | Specification |
| --- | --- |
| Spectral range | 391–1008 nm |
| Spectral resolution FWHM | 0.75–0.82 nm |
| Spatial pixels | 1024 |
| Spectral bands | 784 |
| Detector | CMOS (Complementary metal-oxide- semiconductor sensors) |
| Radiometric resolution | 12 bit |
| Frame rate | 10 fps |
| Output data format | Binary BIL data with separate ASCII format header, Envi compatible |
| Instrument calibration | Spectral calibration. Normalization using internal referencing |

The images were selected, avoiding the shadow and the edge, as shown in Figure 2. The image of $204 \times 177$ pixels of the VNIR corresponds to an $84 \times 73 \ mm^2$ area with a spatial resolution of 2.4 mm/pixels.

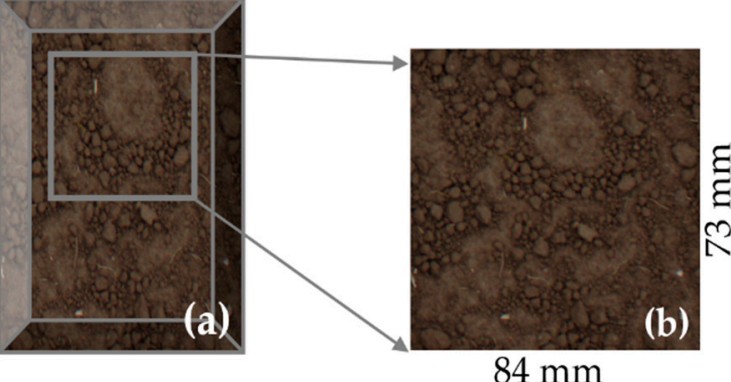

**Figure 2.** Example of an image selected for analysis. In order to avoid shadow, the VNIR image was selected from the upper (**a**) part of the tray. The image subset of $84 \times 73$ mm$^2$ with a pixel size of 2.4 mm/pixels (**b**) was obtained.

### 2.3. Image Processing

Although the spectroscopy data provide useful information for the identification of various materials with similar spectral properties, they suffer from highly correlated and noisy spectral bands [25,26]. Therefore, it is necessary to perform image processing before further data analysis. The noisy bands ranging from 391 nm to 420 nm and 951 nm to 1008 nm of the VNIR sensor were discarded due to a low signal to noise ratio [27]. Although the images were spectrally resized, their dimensionality was high (676 bands). The processing of the enormous amount of the hyperspectral data might be problematic, leading to high computational cost. To decrease the amount of redundant and correlated spectral bands without losing useful information, we used the spectral binning by averaging every four adjacent bands, i.e., binning was set to four (see Section 2.4). As a result, the spectral dimensionality reduced from 676 to 169 (bands).

Various Soil properties affect soil's spectral reflectance, such as Soil particle distribution, OM, Soil moisture, iron oxide, Soil minerals, and Soil roughness. Therefore, we performed the Gaussian stretching, which is similar to the histogram equalization. The idea was to achieve a stretched brightness value distribution, which resembles a normal distribution where the tails were clipped to ±2 standard deviations. The image-processing tools based on the multivariate techniques, such as the principal component analysis (PCA), are often applied to reduce the noise. The PCA involves a linear decomposition of the original dataset into a new coordinate system based on the eigenvectors and principal components (PC) [28]. There can be as many PCs as the number of spectral bands in the original image. However, the first two PCs contain the highest spectral and spatial data variability [29]. The rest of the PCs obtain mostly noise (useless information). Implementing the forward PC rotation, the first two PCs that contained 99% of the data variability were selected [30]. However, the PC images do not allow identification and quantification of the Soil mineralogical changes. Therefore, the transformation of the PC images back into their original data space was completed. Figure 3 shows the flowchart of the image processing.

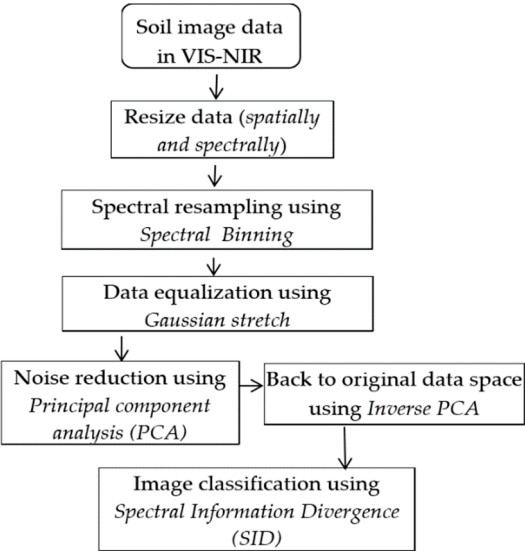

**Figure 3.** Flowchart of the image processing steps followed for each Soil image before Spectral Information Divergence image classification was performed.

### 2.4. Spectral Information Divergence Approach (SID)

The key to imaging spectroscopy classification is the assessment of the spectral similarity of various objects. An image pixel is usually a mixture of different materials with various abundance fractions. Therefore, the high spectral resolution of the sensor (hundreds of spectral bands) does allow resolving these mixtures better than the low spectral resolution. Consequently, the spectral information provided by this pixel is essential for material discrimination, detection, identification and classification [31,32]. Since our goal was identifying and quantifying the Soil surface mineralogy under D-FC, FC and W-FC treatments, the Spectral Information Divergence (SID) spectral similarity approach was used. The SID stochastic classifier provides more accurate results compared to empirical methods [33,34]. The SID uses a divergence measure to match pixels spectra to reference spectra. The more pixels are similar, the smaller the divergence. Pixels are not classified when the divergence measure is greater than the specified maximum divergence threshold. The SID measures the spectral variability of a single mixed pixel where each pixel is considered as a random variable and uses its spectral histogram to define a probability distribution [35]. Considering two spectral vectors, spectral reference r = (r$_1$, r$_2$ ... r$_N$) and an unknown spectral image u = (u$_1$, u$_2$ ... u$_N$) the SID is calculated based on relative entropy. Thus,

$$\text{SID (r, u)} = D\,(r \parallel u) + D\,(u \parallel r), \tag{1}$$

where

$$D\,(r \parallel u) = -\sum_{i=1}^{N} pilog(\frac{pi}{qi}),\ D\,(u \parallel r) = -\sum_{i=1}^{N} qilog(\frac{qi}{pi}), \tag{2}$$

and

$$pi\ =\ ri/\sum_{i=1}^{N} ri,\ qi =\ ui\ /\ \sum_{i=1}^{N} ui, \tag{3}$$

N is the number of bands, the symbol ∥ represents both the relative entropy of u with respect to r and the relative entropy of r with respect to u.

The spectral reference for the SID classifier was selected from the United States Geological Survey (USGS) spectral library [36] based on the X-ray diffraction analysis of our soils, more precisely on the clay fraction of these soils. The clay fraction was determined with the pipette method. The X-ray analysis identified a range of minerals such as goethite, hematite, maghemite, Mg-clinochlore, ferroan

clinochlore, kaolinite, and muscovite. Likewise, the Analytical Spectral Device (ASD Boulder, USA) spectroradiometer acquiring data in the 350 nm to 2500 nm spectral range with a spectral interval of 1.4 nm in VNIR (350–1000 nm) was used to measure the spectra of Soil 2. These measurements were performed under controlled dark conditions, using a 512-channel silicon photodiode array. After the white reference optimization, an average spectrum of Soil 2 was obtained. Besides the spectral information, the SID approach requires a maximum divergence threshold. The image of Soil 1 acquired at the beginning of the experiment was used to determine this threshold (see Table 1). We chose this Soil image because its area covered by hematite was known. By trial and error, the maximum divergence threshold value of 0.15 identified the hematite over the image, as shown in Figure 3. We assumed that this threshold (0.15) could identify the other minerals existing in any Soil image. Therefore, the SID classification was performed on all the Soil samples using this threshold. Together with the threshold trials, various spectral binning for spectral resampling were used when performing the SID classification.

Since each of the SID class had isolated unclassified pixels, a majority post-classification image analysis using a window $3 \times 3$ pixels was carried out. Next, the percentage of each mineral in each image was calculated based on the pixel count in the image. Here, the number of 36,108 pixels corresponded to 100% of the image in VNIR. Therefore, based on the number of pixels that each mineral occupied in the image, its percentage could be determined. It is essential to point out that the percentage of these minerals was derived from surface measurements, and the Soil depth was not considered.

Both the ASD and the USGS spectral information were used independently as input to the SID classifier to detect the hematite in Soil 1 (Figure 4). The percentage of hematite over the Soil image was 36% and 37% for the ASD and the USGS spectral data, respectively. Since this difference was 1%, the USGS spectral library data could be used instead of measured data to monitor the Soil mineralogical changes in VNIR.

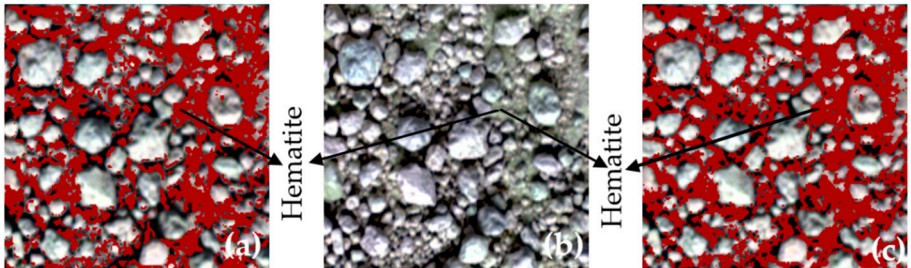

**Figure 4.** The original image (**b**) of Soil 1 at the beginning of the experiment is in the middle. Using Spectral Information Divergence (SID) classifier with a threshold value of 0.15, hematite (cherry color) occurrence over the image was defined. The image classification results, on the left (**a**) and the right side (**c**), were obtained using the Analytical Spectral Device (ASD) and the USGS spectral data, respectively. The black arrows indicate the hematite in the original image and its classification results for both the ASD and the USGS spectral data used.

*2.5. Inductivity Coupled Plasma-Optical Emission Spectrometry Instruments (ICP-OES) Measurements*

The Inductivity Coupled Plasma-Optical Emission Spectrometry Instruments (ICP-OES) technique measures the concentration of chemical elements in a solution applying a linear approach (element concentration vs. light intensity) [37]. To investigate the losses of soluble elements, we collected the excess water of each Soil sample in tubes of 15 mL. Likewise, the samples were acidified with nitric acid to keep the elements in the solution. While the samples were measured every week in the D-FC and the W-FC treatments, these measurements performed every three days in the FC treatment. Using the ICP-OES technique, the concentration of Mg, Al, Ca, K, and Na was determined.

## 3. Results

Applying the USGS spectral library, the minerals identified in the VNIR using the SID classifier belong to the phyllosilicate group more specifically to chlorite, iron oxide and hydroxide. Mg-clinochlore, goethite with average grain size 125 μm, quartz coated 50% by goethite (Qz-Gt) and averaged hematite dimorphous with maghemite (Hm-Mh) with varies grain size intervals (150–250, 60–104, 30–45, 10–20 μm) were used to investigate the mineral variations over time. Furthermore, the ferroan clinochlore observed at the dry conditions was not considered for further analyses. The image processing and analyses were carried out for each Soil sample determining the percentage of minerals. Next, we summarized the mineral percentages by averaging triplicates of each Soil type (Soil 1–Soil 3) placed in each Soil treatment (D-FC, FC, and W-FC). Figure 5 shows the results at the D-FC Soil treatment at the start (week 0), middle (week 4), and the end of the experiment (week 8) in the dry conditions. All the soils showed changes in their minerals distribution over time. One of the Soil properties that affect the Soil spectral reflectance is indeed the OM. However, it is essential to specify the species of the OM occurring in the Soil because their composition determines the spectral region that they are active. The OM of Soil 3 comes from the sewage sludge manure. The main components in the sewage sludge are lignin and cellulose, which have absorption features in the SWIR. Therefore, in the VNIR, the sewage sludge manure does not affect the spectral reflectance of the minerals.

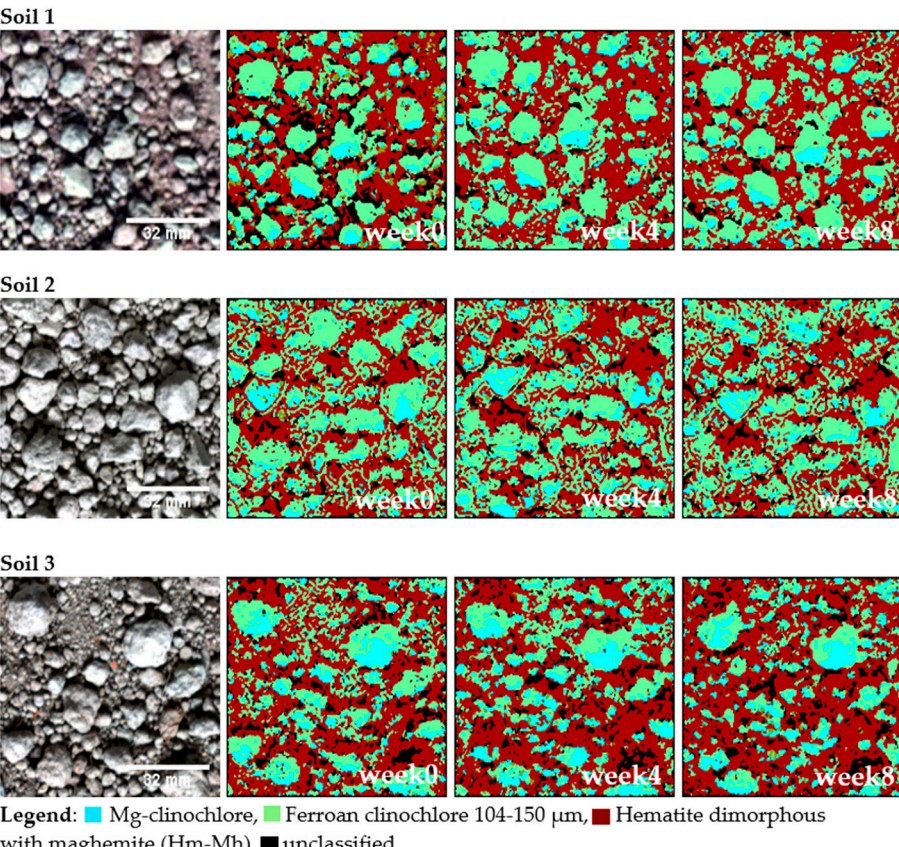

**Figure 5.** Example of the SID classification results at the drying-field capacity (D-FC) treatment at the start (week 0), middle (week 4) and the end of the experiment (week 8) in the dry conditions (Soil 1–Soil 3). The colors represent the minerals identified in the VNIR. All the soils show changes in their mineral distribution over time. The original images at the start of the experiment, together with the scale bar, are also shown.

### 3.1. Drying-Field Capacity Treatment (D-FC)

The Mg-clinochlore could be considered stable over time regardless of the Soil conditions (wet or dry) or the presence of OM (Figure 6a). However, in the wet conditions, the percentage of Mg-clinochlore of Soil 2 and Soil 3 was significantly higher compared to Soil 1.

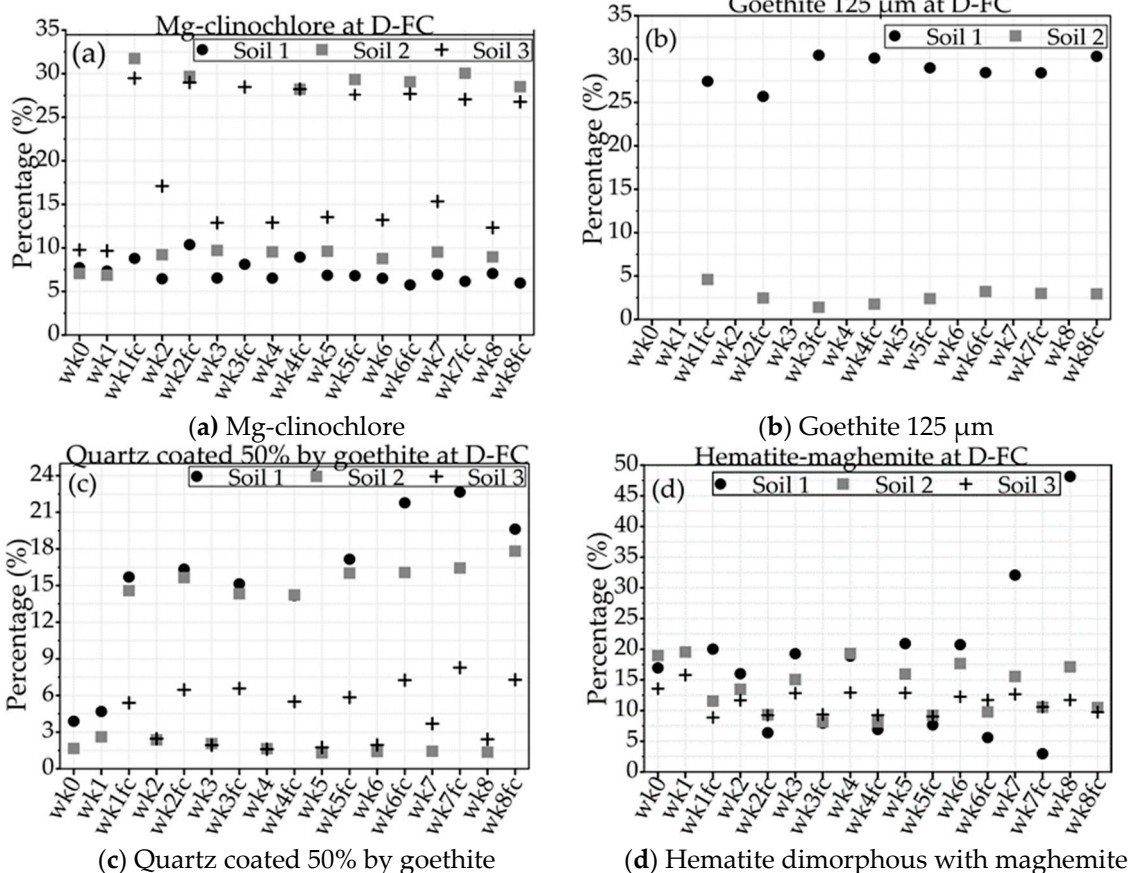

(**a**) Mg-clinochlore

(**b**) Goethite 125 μm

(**c**) Quartz coated 50% by goethite

(**d**) Hematite dimorphous with maghemite

**Figure 6.** The average percentage of Mg-clinochlore (**a**), goethite 125 μm (**b**), quartz coated 50% by goethite (**c**) and hematite dimorphous with maghemite (**d**) changes for all the soils at the D-FC treatment. The circle, square and plus symbols represent Soil 1, Soil 2, and Soil 3, respectively. The vertical axis characterizes the percentage of each mineral occurring in an image. The scale of the Y-axis varies from 0% to 100%. In the horizontal axis, wk1, ..., wk8 stands for week 1, ..., week 8, when the Soil sample was at the dry conditions. Since the Soil samples were at the field capacity every three days, wk1fc, ..., wk8fc (week 1fc, ..., week 8fc) was used to represent it.

The goethite 125 μm was not detected for Soil 3 (high OM) at the D-FC Soil treatment (Figure 6b). In the wet conditions, the percentage of goethite of Soil 1 was significantly higher compared to Soil 2. Although the percentage of goethite fluctuated with a small magnitude over time, it tended to increase and decrease for Soil 1 and Soil 2, respectively, at the end of the experiment.

The Qz-Gt of Soil 1 was observed only in the wet conditions after the first week, and its percentage increased in the second half of the experiment (Figure 6c). Likewise, at the wet state, the Qz-Gt of Soil 2 and Soil 3 increased with two percent at the end compared to the start of the experiment.

The Hm-Mh was stable at the wet conditions for all the soils. While at the dry conditions, the Hm-Mh of Soil 1 increased significantly after week six, this mineral fluctuated over time for Soil 2 and Soil 3 (Figure 6d). The magnitude of these fluctuations was higher for Soil 2 compared to Soil 3.

### 3.2. Field Capacity Treatment (FC)

The Mg-clinochlore of Soil 1 increased until week four, and after week six, it settled for the rest of the experiment (Figure 7a). On the other hand, the Mg-clinochlore of Soil 2 and Soil 3 decreased continuously after week two until the end of the experiment.

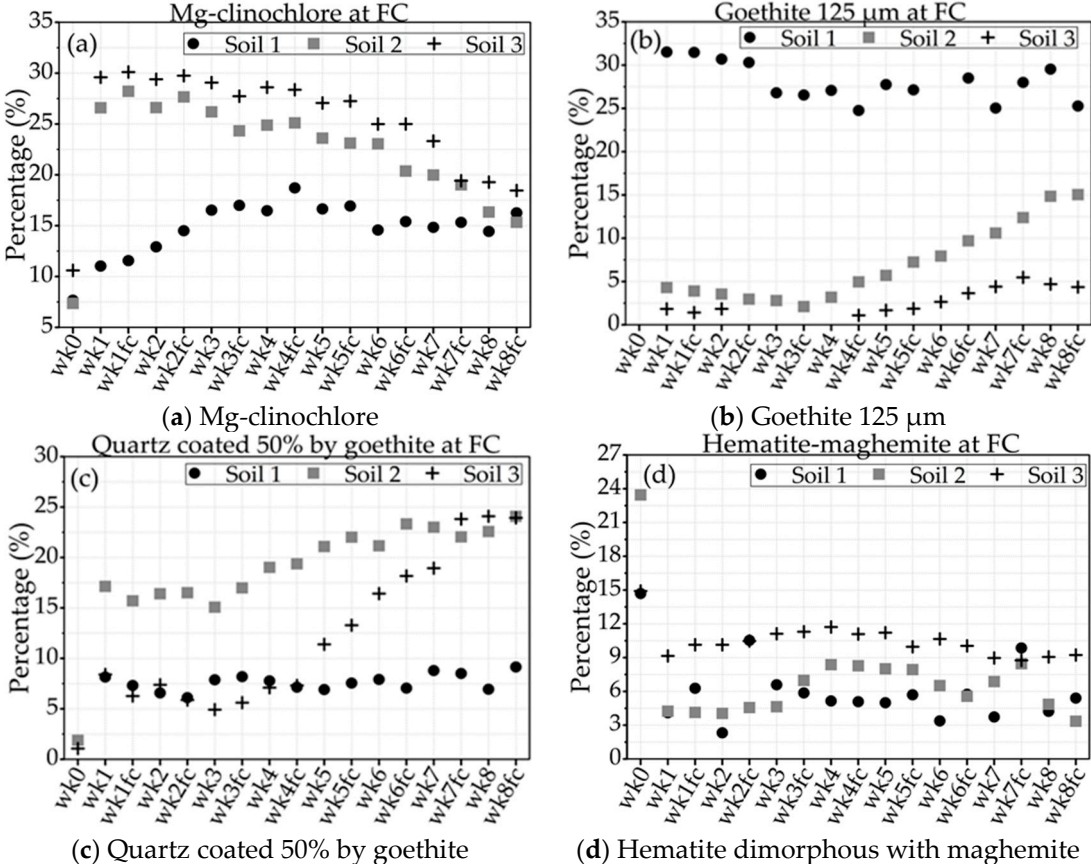

(**a**) Mg-clinochlore　　　　　　　　　　　　　　　　(**b**) Goethite 125 μm

(**c**) Quartz coated 50% by goethite　　　　　　　　(**d**) Hematite dimorphous with maghemite

**Figure 7.** The average percentage of Mg-clinochlore (**a**), goethite 125 μm (**b**), quartz coated 50% by goethite (**c**) and hematite dimorphous with maghemite (**d**) changes for all the soils at the FC treatment. The circle, square and plus symbols represent Soil 1, Soil 2 and Soil 3, respectively. The vertical axis characterizes the percentage of each mineral occurring in an image. The scale of the Y-axis varies from 0% to 100%. In the horizontal axis, wk1, ..., wk8 stands for week 1, ..., week 8. Since the experiment was performed every three days, wk1fc, ..., wk8fc (week 1fc, ..., week 8fc) was also used to represent the results. Here, the Soil samples were at the field capacity all the time.

The goethite of Soil 1 decreased for the first half of the experiment. Although this mineral fluctuated in the second half, its percentage was lower compared to the start of the experiment. The goethite of Soil 2 and Soil 3 increased in the second half of the experiment. However, the goethite of Soil 3 settled in the last two weeks of the experiment (Figure 7b).

The Qz-Gt of Soil 1 was stable for the entire time that the experiment ran. For Soil 2 and Soil 3, this mineral increased after week three (Figure 7c). While the Qz-Gt of Soil 2 stabilized after week six, this mineral settled in the last week of the experiment for Soil 3.

The first FC treatment led to the Hm-Mh decrease for all the soils (Figure 7d). However, this decrease in the percentage of Hm-Mh was higher for Soil 2 compared to Soil 1 and Soil 3. After the first week, the Hm-Mh of Soil 3 and Soil 1 (except some variations) were stable for the rest of the experiment. Although the Hm-Mh of Soil 2 fluctuated over time, its percentage was similar in the first week and the end of the experiment.

### 3.3. Waterlogging-Field Capacity Treatment (W-FC)

The Mg-clinochlore increased for all the soils after the first W-FC treatment (Figure 8a). Moreover, the Mg-clinochlore of Soil 1 increased until week three. After that, this mineral stabilized for Soil 1 for the rest of the experiment. On the other hand, the Mg-clinochlore of Soil 2 and Soil 3 decreased continuously after week five until the end of the experiment. However, this decrease was more noticeable for Soil 2 compared to Soil 3.

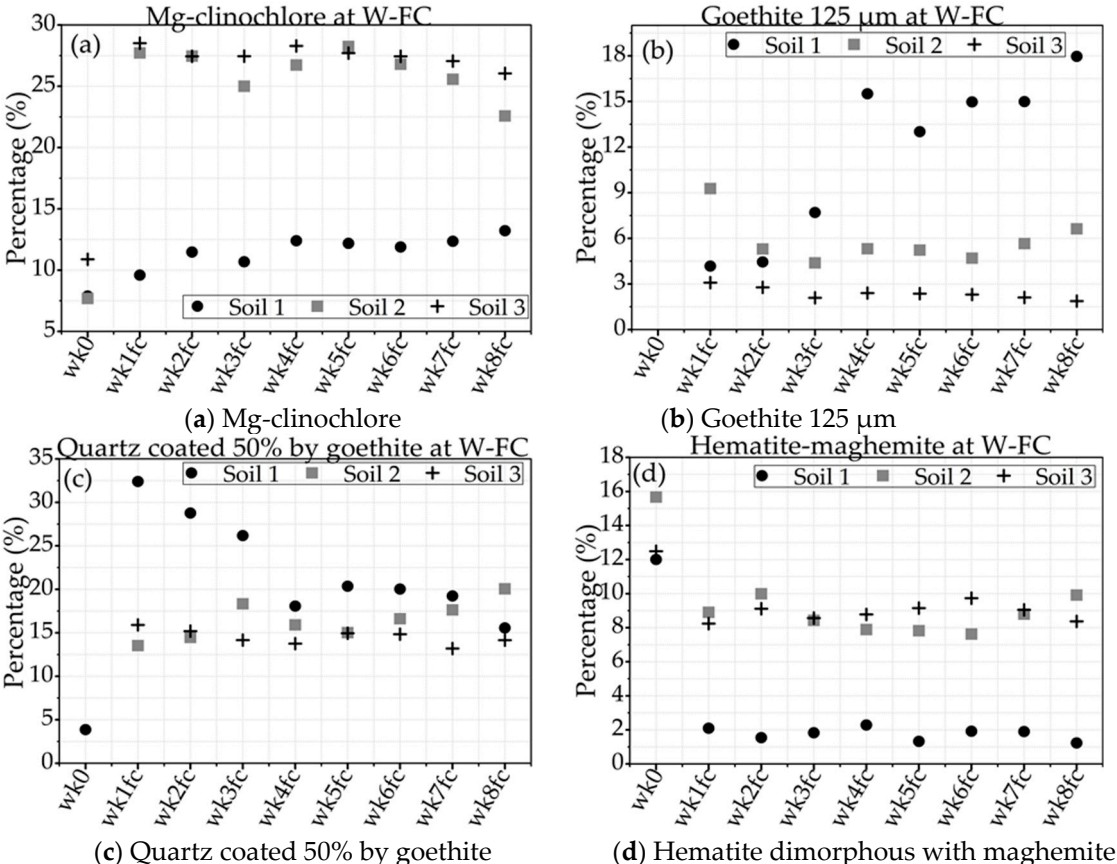

(**a**) Mg-clinochlore  (**b**) Goethite 125 μm

(**c**) Quartz coated 50% by goethite  (**d**) Hematite dimorphous with maghemite

**Figure 8.** The average percentage of Mg-clinochlore (**a**), goethite 125 μm (**b**), quartz coated 50% by goethite (**c**) and hematite dimorphous with maghemite (**d**) changes for all the soils at the W-FC treatment. The circle, square and plus symbols represent Soil 1, Soil 2, and Soil 3, respectively. The vertical axis characterizes the percentage of each mineral occurring in an image. The scale of the Y-axis varies from 0% to 100%. In the horizontal axis, wk1fc, ..., wk8fc stands for week 1fc, ..., week 8fc when the Soil samples were at the field capacity conditions. There was no data available at the waterlogging conditions.

The goethite 125 μm of Soil 1 increased continuously until the end of the experiment. After the decrease of the first week, the goethite of Soil 2 stayed stable for the rest of the experiment. This stability was also observed for the goethite of Soil 3 (Figure 8b).

The Qz-Gt of Soil 1 decreased continuously for the entire period that the experiment ran. Although with a small magnitude, the Qz-Gt of Soil 2 tended to increase over time. Differently, the Qz-Gt of Soil 3 stayed more or less stable for the entire period of the experiment (Figure 8c).

Figure 8d shows that the Hm-Mh dropped considerably after the first waterlogging treatment for all the soils. However, this decrease of Hm-Mh was more notable for Soil 1 compared to Soil 2 and Soil 3. Later, the Hm-Mh remained stable for the rest of the experiment for all the soils.

Figure 9 summarized the percentage of the minerals at the start and end of the experiment when the Soil samples were at field capacity. An exception was the Hm-Mh at the D-FC treatment, where the Soil samples were at the dry conditions.

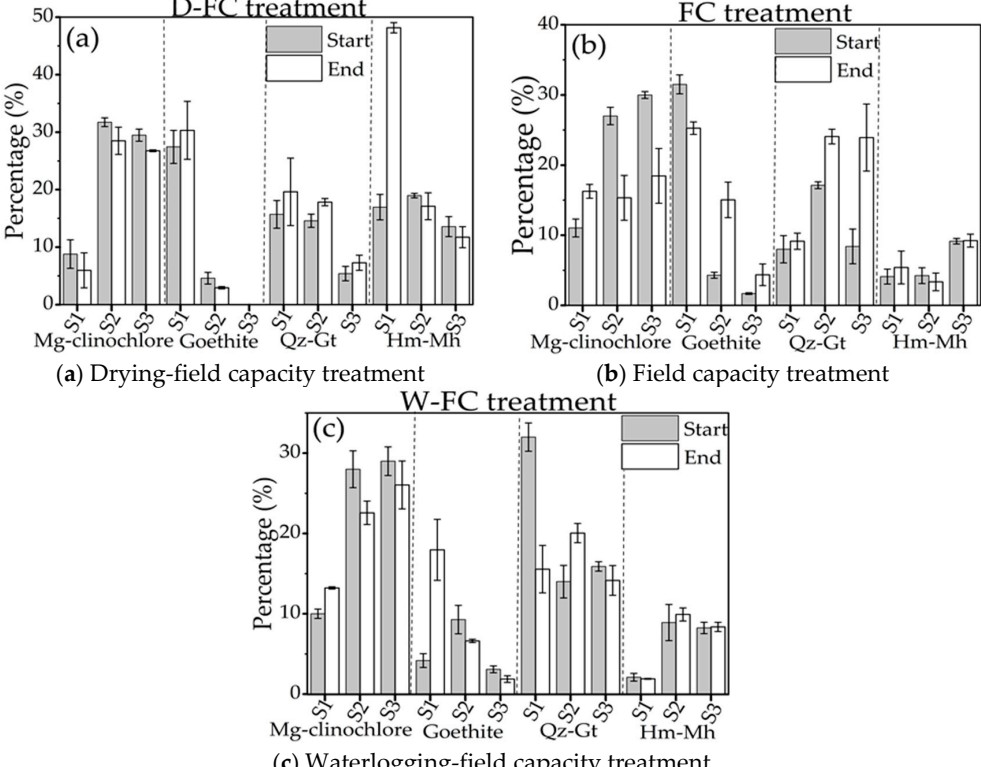

(**a**) Drying-field capacity treatment

(**b**) Field capacity treatment

(**c**) Waterlogging-field capacity treatment

**Figure 9.** The percentage of the minerals when the Soil samples were at the field capacity. An exception was the Hm-Mh at the D-FC treatment, where the Soil samples were at the dry conditions. The vertical axis characterizes the percentage of each mineral occurring in an image. The horizontal axis represents the Mg-clinochlore, goethite 125 μm, quartz coated 50% with goethite (Qz-Gt) and hematite dimorphous with maghemite (Hm-Mh) of Soil 1–Soil 3 (S1–S3) in the drying-field capacity (**a**), field capacity (**b**) and waterlogging- field capacity (**c**) treatments at the start and end of the experiment. The error bars represent the standard deviation of the minerals in triplicated Soil samples. Since the SID approach disregards the self-shadow areas created by various Soil aggregate sizes, these aggregate variations influenced the standard deviation.

At the D-FC treatment, the Mg-clinochlore decreased for all the soils at the end of the experiment. On the other hand, the Qz-Gt increased at the end of the experiment. An increase in the goethite and the Hm-Mh of Soil 1 occurred at the end of the experiment. On the contrary, these minerals (goethite and Hm-Mh) decreased for Soil 2 and Soil 3 at the end of the experiment at the D-FC treatment, as is shown in Figure 9a.

At the FC treatment, the Mg-clinochlore increased and decreased for Soil 1 and Soil 2, Soil 3 at the end of the experiment, respectively. Oppositely, the goethite decreased and increased for Soil 1 and Soil 2, Soil 3 at the end of the experiment, respectively. An increase of the Qz-Gt for all the soils occurred at the end of the experiment. The Hm-Mh increased for Soil 1, decreased for Soil 2, and stayed stable for Soil 3 at the end of the experiment (Figure 9b).

At the W-FC treatment, the Mg-clinochlore and the goethite declined at the end of the experiment for Soil 2 and Soil 3 (Figure 9c). On the contrary, these minerals (Mg-clinochlore and goethite) increased for Soil 1 at the end of the experiment. The other mineral, Qz-Gt, decreased for Soil 1 and Soil 3 and increased for Soil 2 at the end of the experiment. The Hm-Mh of Soil 1 and Soil 3 stayed stable, whereas it increased for Soil 2 at the end of the experiment.

### 3.4. ICP- OES Results

The CEC refers to the ability of negatively charged Soil particles to attract and retain cations such as $Ca^{2+}$, $Mg^{2+}$, $K^+$, $Na^+$, $Al^{3+}$ by the electrostatic forces.

At the D-FC treatment, the loss of Ca, Mg, K, Na, Al of Soil 1 increased at week two (Figure 10a). After that, it decreased continuously over time. However, this decrease was higher for the Ca, followed by K, Mg, Na, and Al. Regardless of the amount of OM, the concentration of cations of Soil 2 and Soil 3 was low and stayed stable over time. Magnesium was not detected for Soil 2 and Soil 3 in the D-FC treatment.

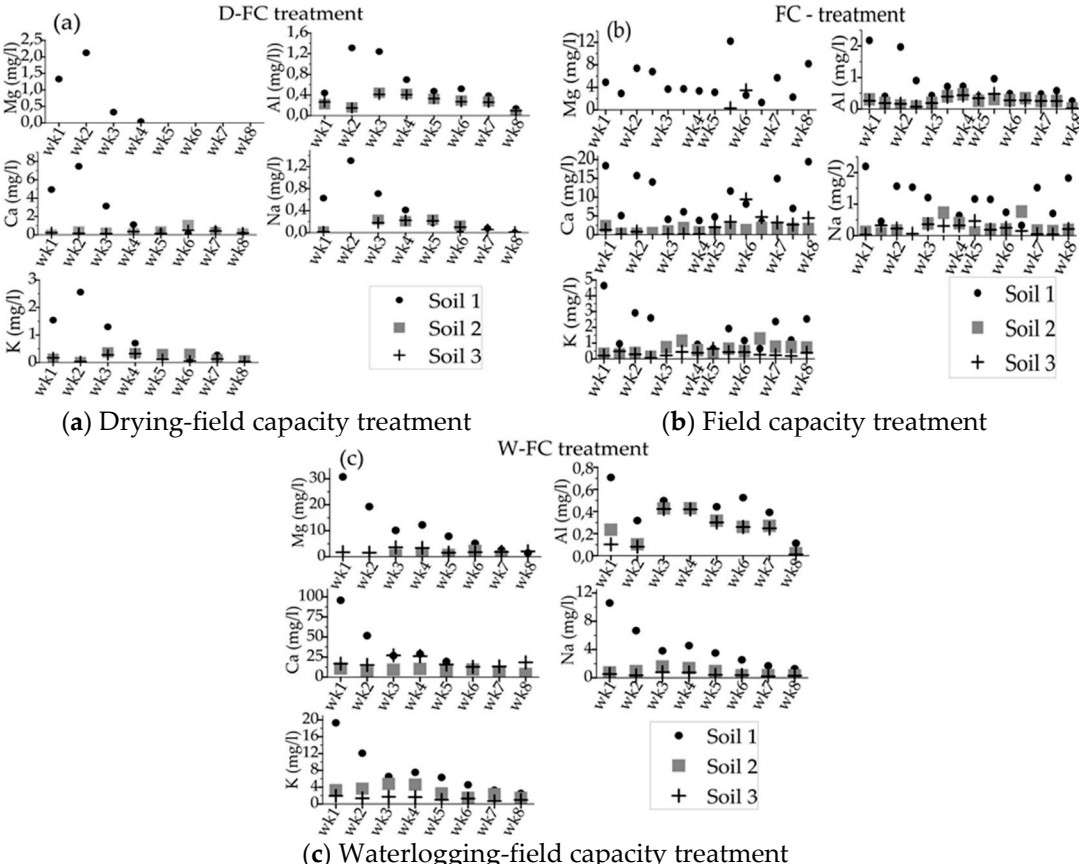

(**a**) Drying-field capacity treatment      (**b**) Field capacity treatment

(**c**) Waterlogging-field capacity treatment

**Figure 10.** The concentration of cations determined using the Inductivity Coupled Plasma-Optical Emission Spectrometry Instruments (ICP-OES) technique at the drying- field capacity (**a**), field capacity (**b**), and waterlogging- field capacity (**c**) treatments for Soil 1–Soil 3. While the results were every week for the D-FC and the W-FC treatments, the results were at three days basis for the FC treatment.

At the FC treatment, the loss of K, Na, and Al of Soil 1 decreased at the end compared to the start of the experiment. Moreover, while the Mg increased at the end of the experiment, the Ca of Soil 1 did not change. The concentration of the elements of Soil 2 and Soil 3 fluctuated with a small magnitude over time, except for the Mg, which was not detected. (Figure 10b).

Figure 10c shows that the loss of Ca, Mg, K and Na of Soil 1 decreased overtime at the W-FC treatment. While this decrease was fast until week three, it slowed down for the rest of the experiment. The other element, Al, decreased at week two to stay stable until week six to drop again until the end of the investigation. The concentration of Ca, Mg, K, and Na of Soil 2 and Soil 3 did not change over time at the W-FC treatment.

## 4. Discussion

### 4.1. Drying-Field Capacity Treatment (D-FC)

Repeated drying and field capacity cycles can cause changes in the Soil chemical composition due to the migration of Soil chemical elements such as Ca, Fe, Mg, Al, K, Na, or Si [38,39]. Moreover, our ICP-OES results indicated that the loss of soluble elements was noticeable for Soil 1. On the contrary, the presence of OM (Soil 2 and Soil 3) diminishes the loss of cations from the Soil matrix. The image analysis showed that the Mg-clinochlore of Soil 1 stayed stable over time regardless of the Soil conditions (dry or field capacity). The Mg-clinochlore of Soil 2 and 3 fluctuated with a small magnitude, considering this mineral relatively stable. This stability of the Mg-clinochlore at the D-FC treatment could be related to the increase of the cohesion between the organic molecules and the mineral surfaces in the dry conditions [40].

The goethite is active at the wet conditions, whereas the hematite prevails in the dry Soil conditions. Indeed, the SID classifier detected the goethite only at the field capacity conditions. This mineral was stable over time. The goethite adheres to the quartz surfaces by covalent Fe-Si-O bonds [41] occurring in our experiment as quartz coated 50% by the goethite. The presence of OM makes the goethite negatively charged, allowing it to precipitate over time.

Moreover, the fulvic and humic acids increase the weathering of quartz, especially when the Soil is at the neutral pH [42,43]. However, our results showed that the Qz-Gt stayed stable over time for the soils with OM. Our results could be explained with the presence of Al and Fe oxides stabilizing the Soil upon drying, as well as the short period (eight weeks) that the experiment ran. Due to the absence of OM, the goethite is positively charged. Therefore, it could be immobilized due to the attractive electrostatic interactions with the negatively charged Soil matrix minerals, such as the quartz [44]. However, our results showed an increase in the Qz-Gt of Soil 1 in the second half of the experiment. These results might be related to the changes in the Soil 1 mineralogy, such as the recrystallization of Fe and Al oxyhydroxides, when exposed to high-temperature [45,46].

Like the other minerals, the Hm-Mh of the soils with OM was relatively stable for the duration of our experiment. Indeed, the Fe oxides have a large specific surface area playing a significant role in forming the organo-mineral associations within soils [47,48]. Also, [10] showed that the interactions between the amorphous iron oxides and the OM were significant to Soil stability. The Hm-Mm of Soil 1 increased at the end of the experiment. This increase could be related to both the hematite and the maghemite. The added hematite in Soil 1 did not have enough time to integrate into the Soil matrix leading to leaching and attachment of the small grain sizes to bigger ones. Moreover, the maghemite is poorly crystallized, leading to its mobility and precipitation in the porous media [49,50].

### 4.2. Field Capacity Treatment (FC)

The Soil moisture controls mainly the chemical weathering and the mineral transformations [51]. When the Soil was at the field capacity conditions, the Mg-clinochlore of Soil 2 and 3 (low and high OM) decreased over time. Indeed, the instability of the chlorite minerals in the Soil is known [52]. This instability could be related to the large molecule of magnesium, which does not stick tightly to the Soil particles. As a result, it could easily leach from the Soil at the neutral pH [53].

Moreover, generally, the OM can stabilize the Soil by increasing the inter-particle cohesion within the Soil aggregates [54]. However, the manure (present in our soils) does not increase the Soil cohesion [55]. Therefore, the leakage of Mg and its weak inter-particle cohesion lead to a decrease in the Mg-clinochlore.

Both the goethite and the Qz-Gt of Soil 2 and 3 increased at the end of our experiment. The Soil at the wet conditions is not fully oxic, which may lead to reducing conditions. Under such conditions, the accumulation of crystalline Fe hydroxide phases such as the goethite might occur [56,57].

The associations of goethite with the OM is weak because the coverage of goethite by the OM prevents the action of the attractive forces, and therefore, increasing the electrostatic repulsive forces

between colloids [58]. The precipitation of goethite and weathering of the quartz simulated by the fulvic and humic acids could lead to the accumulation of the Qz-Gt over time.

Regardless of the Soil type, the Hm-Mh was relatively stable over time. Both the crystalline or amorphous Fe species (hematite or maghemite) might interact with the Soil inorganic and organic components due to van der Waals attraction [59]. These results could be related to the hematite suppression at the field capacity conditions.

The Mg-clinochlore, goethite, and Qz-Gt of Soil 1 were stable. One explanation is that Soil 1 went through the high temperature, declining the exchangeable cations, and making this Soil in an unchanging condition [60].

In the Soil environment, where oxygen is sufficient, the attachment of the aerobic microorganisms on the mineral surface is a known process [61]. As a result, the attached microorganisms dissolve the minerals based on their nutritional needs causing mineral dissolution and the element mobilization such as Fe, Mg, Al and Si [62]. Therefore, the minerals variations in the FC state could be related to the mineral weathering by the microorganisms.

### 4.3. Waterlogging-Field Capacity Treatment (W-FC)

Our experiment showed that the first waterlogging-field capacity treatment had a significant effect on the Soil mineralogy, regardless of the Soil type. However, for the rest of the experiment, the minerals were stable, especially for the soils with OM.

The Mg-clinochlore was stable over time. The minerals dissolved in the waterlogged soils, and consequently, the elements such as Mg can be released into the solution at a low redox potential (waterlogging state) [63,64]. Our ISP-OES results also showed the loss of Ca, Mg, K and Na of Soil 1. However, the image analysis suggested that the Mg-clinochlore of Soil 1 stayed stable. These results indicate the importance of considering the changes in the minerals together with other Soil components and not per se because they depend on the interactions with the other soil components.

The OM interactions with the Soil minerals by the cationic bridges are considered weak. Moreover, the OM adsorbed on the mineral surfaces does not cover the whole particle surface, but it forms patches [65,66]. Even the newly adsorbed OM binds to the existing patches and not to the free mineral surfaces [67]. Therefore, the Mg and the OM might not be capable of creating the new bridges under W-FC treatment, causing the stability of the Mg-clinochlore for Soil 2 and Soil 3.

Although the goethite can precipitate from the solution at the normal environmental pH, the presence of Al prevents this process [68]. Likewise, it was proved by [69] that the nanocrystalline goethite transformed into the microcrystalline hematite and goethite as a result of the Soil redox fluctuations. The anaerobic Soil conditions could cause the dissolution of the maghemite [70,71]. Also, it was found that Al destroys the vacancy ordering in the maghemite [72]. However, in the waterlogging conditions, the nanoparticles could remain in the suspension, or they might aggregate and precipitate from the solution. During aggregation, dissolved ions stay within the aggregates due to the inter-particle pore space leading to the aggregate settling.

Moreover, at the neutral pH, the unstable particles dissolve and re-precipitate on the surface of other growing particles [73]. Therefore, the stability of the goethite, Qz-Gt, and Hm-Mh could be related to the re-precipitation process. Moreover, in the oxic conditions, the Fe precipitates as goethite and hematite from the silicate minerals. Due to the Si and Al or OM, the $Fe^{3+}$ and $Fe^{2+}$ oxyhydroxides have a low reducing ability over time [74,75]. The increase of the goethite and the decrease Qz-Gt of Soil 1 could be related to the added hematite manually.

The high clay content increases the Soil cohesion and decreases permeability for the dissolved and colloidal compounds making the leaching process less pronounced. Therefore, high clay content has a positive influence on Soil stability. Soil 3 with 23% kaolinite showed fewer variations at the W-FC treatment than the Soil 2 with 16% kaolinite. Of course, the clay content and the clay mineralogy have a significant effect on the Soil organo-minerals interactions. Therefore, in the soils dominated by 1:1 clay, the aggregation occurs due to the binding capacity of the minerals themselves. Instead, the clay

forms bridges with the polyvalent metal and the OM in the soils with a 2:1 layer structure [76]. We did not investigate the effect of kaolinite because this clay mineral is detected in the SWIR region of the electromagnetic spectrum.

The SID classifier is a probabilistic approach that uses the divergence measure to compare each pixel spectra with the reference spectra. If this divergence, which is related to a threshold, is small, then the pixel spectra are close to the reference spectra. Hence, the pixels that have the divergence greater than the specified threshold are not classified. The threshold value used in this study was 0.15, but it could vary for different mineral types. A threshold that discriminates well one mineral might be either too sensitive or not sensitive enough for another mineral due to the similar/dissimilar nature of their probability distributions. Besides unclassified pixels, misclassification due to a false positive or false negative errors could occur as well. Therefore, the threshold used in the SID classifier should be specific for each mineral. At the start of the experiment, in the D-FC treatment, the percentage of unclassified pixels was 14%, 13%, and 14% for Soil 1, Soil 2, and Soil 3, respectively. When the Soil was in wet conditions, the percentage of unclassified pixels was 29%, 25% and 24% for Soil 1, Soil 2 and Soil 3, respectively.

One explanation might be related to the behavior of the minerals, such as the ferroan clinochlore, which was observed only in the dry conditions (data not shown). Likewise, the hematite prevails in the dry conditions, and the goethite overcomes in the wet Soil conditions. Furthermore, the Soil aggregates lead to self-shadowing of the Soil surface resulting in a low reflectance value. The SID classifier characterizes each mineral type by using its corresponding spectral signature, neglecting the spectral information associated with the shadow. Since the shadow area depends on the Soil aggregates size and their occurrence in the image, the percentage of unclassified pixels varies for each Soil sample. However, [35] pointed out that the SID classier is insensitive to the illumination effects and the brightness shifts. Although the dimensionality of the images was reduced based on the added hematite (Section 2.4), a large number of available spectral bands (169) and the small training samples (8) could have affected the classification accuracy known as the Hughes phenomenon [77].

The RS classification approach requires the validation of the results with the ground truth data. Unfortunately, the ground truth information is not available all of the time. In this study, we used the spectra from the ASD and the USGS to define and quantify the distribution of the hematite added manually in the Soil image. Since the hematite percentage differed with 1% when using the ASD and the USGS, then the latter could be used instead of measured data. The occurrence of the other minerals in the Soil was unknown. Therefore, we assumed that the USGS spectral library could be used to capture the distribution of the minerals occurring in the soil.

The Soil organo-mineral associations occur over a range of bonding mechanisms such as the cation and anion exchange, the water and cation bridging, the ligand, the hydrogen, and van der Waals and the hydrophobic interaction [12]. While some of these organo-mineral bonds are assumed to be susceptible to disruption, others are considered stable, therefore, affecting the Soil stability in different ways. Hence, it is crucial to know their behavior at the high spatial and temporal resolution as well at the vertical Soil horizons. RS plays a significant role in acquiring relevant data. However, at this stage, it is difficult to capture in detail the Soil organic-mineral interactions, which occur at the nanoscale, to field-scale observations. Likewise, the Soil organo-mineral bonds at the subsurface level with RS need attention. At this point, potential users can be the scientific community dealing with Soil stability and environmental concern.

## 5. Conclusions

In this study, the SID image classifier derived from the VNIR spectroscopy data was used to monitor the Soil surface mineralogical alterations under laboratory conditions. We chose three Silty Loam soils varying in the OM content (no, low and high). The trays with aggregates were exposed to the D-FC, FC and W-FC treatments. These treatments were performed because there is limited knowledge of the Soil mineralogical behavior at different moisture conditions occurring in a short

period. The ICP-OES measurements were carried out to quantify the losses of the soluble elements by percolation for each soil sample.

The assumption was that using the VNIR spectroscopy data changes in the Soil surface mineralogy due to the moisture variations could be detected and quantified over time. Using the SID approach, the image area occupied by the Mg-clinochlore, goethite, quartz coated 50% by goethite, hematite dimorphous with maghemite was determined. Our results showed that these minerals behaved differently, depending on the Soil type and Soil treatment. While for the soils with OM, the mineralogical alterations were evident at field capacity state, for the one with no OM, these changes were insignificant. However, regardless of the Soil type, these minerals were stable at the D-FC and W-FC treatments. Using imaging spectroscopy data on the Silty Loam soil, we showed that the surface mineralogy changes over time, depending on the Soil type and the moisture conditions. Changes in the minerals composition might cause changes in the Soil aggregate stability. Therefore, it will be interesting to investigate the effect of these mineralogical changes on Soil aggregate stability.

**Author Contributions:** Conceptualization, I.Y. and F.v.d.M.; methodology, I.Y.; validation, I.Y.; formal analysis, I.Y.; investigation, I.Y.; data curation, I.Y.; writing—original draft preparation, I.Y.; writing—review and editing, D.P.S. and F.v.d.M.; supervision, D.P.S. and F.v.d.M.

**Funding:** This research received no external funding.

**Acknowledgments:** The authors would like to thank the colleagues at the Hogedruklab (University of Twente) for using their facilities to produce the Soil sample without organic matter. We also thank our laboratory colleagues at the Faculty of Geo-Information Science and Earth Observation (University of Twente) for arranging the place and helping during the experimental setup.

**Conflicts of Interest:** The authors declare no conflict of interest.

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
