# Peer review of "Monitoring Soil Surface Mineralogy at Different Moisture Conditions Using Visible Near-Infrared Spectroscopy Data"

_remotesensing, doi:10.3390/rs11212526_

Round 1

Reviewer 1 Report

The article deals with the monitoring of alterations of soil surface mineralogy due to different moisture conditions with vis-nir spectral data.

Mapping of soil surface mineralogy with remote sensing data is very difficult but important task.

Presented manuscript presents very interesting results but lacks in their proper analysis and explanation.

First of all, as the analysis is performed for soil surface it should be mentioned more clearly starting from the title of the article.

The focus of the introduction should be moved more to soil surface mineralogy, the reasons and the magnitude of the changes and their detection with remote sensing data. Because mineralogy is the primary focus of the paper not soil aggregation.

Some specific questions are listed below:

Line 123, 129, 136 – how the water was added, how its amount was determined for each treatment for each soil. How is it possible to uniformly distribute 20 ml of water on the surface of soil sample?

Line 143 – how were soil samples covered? What material was used? Did this cover create anaerobic conditions which might have caused some of the observed transformations?

Line 159 – some word seems missing

Line 259 –Did you perform X-ray analysis for studied soils? If so, please, provide more detailed information. Anyway, this point needs further clarification.

Figure 4 –Organic matter is known to mask spectral signal of soil minerals as it coats their grains. Thus when soil has high organic matter content (like soil 3) it means that most part grains of minerals (almost all) are coated by organic matter. Therefore minerals in such soils did not contribute to soil spectral reflectance or their contribution is very limited, which makes it almost impossible to map them based on surface spectral reflectance. So in case of soil 3 which has 12.3 % of organic matter, the largest area should fall into class Unclassified. Could you explain how it was possible to identify such large areas of minerals on soils 3 and 2?

It is also not clear why Soil 3 initially has the area covered by hematite close to soil 1 (where organic matter was removed and hematite was added). Please, explain this.  

Figure 5-7 what is the meaning for wk0, wk1, wk1fc,.. etc?

Figure 8 – I don’t think it is correct to summarize like that. It gives the wrong impression on some data. For example, in case of dynamics of Mg-clinochlore in treatment D-FC for soil 2 and 3, if you end at wk8 (or 7) there will be no change by the end of experiment, but if you end at wk8fc, its area increased.

Line 453 – Did you study soil aggregation process during your experiment? If you refer to it, explain properly why it changes and how.  Actually your discussion looks more like a review. Because you do not relate the results you obtained to the references you mention.

Line 472 – Have you measured pH of studied soils prior to and during (or at least at the end of) experiment? Agricultural soils do not always have pH 7.

I will advise on rewriting the discussion.

First of all, pay attention to the magnitude of the changes. Soil mineralogy is rather stable characteristic. Which of the changes that you observed are the real changes? And not the increased or decreased masking influence of organic matter. Could you compare your results on the magnitude of changes in mineralogy with other studies? Which changes lie within the error range of your classification method? Which changes can be caused by the design of experiment (temperature, coverage). Be a bit more critical to your results.

Try to describe the mechanism of observed changes step by step. Then it will be easier to understand your explanations.

Try to explain your results and not just provide some references or useful information. Try to compare your results with similar research.

Moreover, you should discuss the meaning of your results for the remote sensing, further directions of the research. What is the practical value of your results? How reliable is presented method of remote detection of surface mineralogy?

Author Response

Dear Reviewer, 

We want to thank you for your comments, which improved the manuscript.

Kind regards,

Irena Ymeti and co-authors

Reviewer 2 Report

The presented manuscript reports the results of experiments conducted in order to evaluate the mineralogical changes of in soil samples due to moisture and organic matter variations by means of VNIR imaging spectroscopy data.

Though not extensive, the manuscript has repetitive text and requires English language editing for structure and wording. Also, revising the flow of the text will be beneficial for a better read and clarity.

Below I have some comments and suggestings to be considered:

Ln30: The sentence could benefit form revision for clarity. For instance, in its present form it reads as if climate and land management practices are considered as soil properties.

Ln85: It is not clear what is the "goal" here; a goal has yet to be set. Referring to the previous sentence and the flow "To achieve our goal" could be replaced by " To test our hypothesis". Also, as the data collection was performed using a scanner, "scanned" rather than"photographed" would be the more apt term to be used throughout the manuscript.

Ln86: Please consider replacing "without" with "no". Also, "drying-wetting" and "drying-field capacity" phrases seem to be used interchangeably in the manuscript please consider using one of them for consistency.

Ln95: "...with low and high OM content."

Ln104: " in this study" should be deleted.

Table1: Properties of Soil1 are not aligned. The notation for decimals should be with a period not a comma.

Ln123: is 20mL right? In the next paragraph the amount is reported as one order of magnitude higher.

Ln147: A brief explanation as to why they were "useless" could be "useful" - please consider revising as it sounds colloquial

Scheme1: I suspect the scheme is not an adequate representation of the setup, please consider revising the scheme or better yet please consider including an image of the setup in lieu of this scheme.

Ln161: pushbroom scanner

Table2: Please check text alignment.

Ln166-174: Scheme1 and Table2 summarize most of the info here and so, some text can be discarded for brevity and clarity.

Ln194 and Figure2: It is not clear to me why data normalization has been performed. If the pixel values are of reflectance these values, in theory, are already normalized. If the stretch is applied for visual display, it will not affect the data values for further analysis. Please clarify this.

Ln199: PCA is a well-known and widely used approach and so, this section can be shortened significantly focusing only what has been done in the context of this study and be included with the previous section.

Figure2: While spectral binning does indeed reduce dimensionality of HSI, I believe a more apt term would be spectral resampling rather than dimensionality reduction.

Ln237: This statement is not necessarily accurate: the mixture of different materials in a pixel is a physical phenomenon and independent of the number of spectral bands of the used sensor (or subsequent image). High spectral resolution of the sensor does allow resolving these mixtures better than low spectral resolution.

Ln242: What are these groups of spectral similarity methods? Please consider revising the sentence for clarity.

Ln260-266: While this info could be useful to understand the formation of minerals, it is not relevant to the SID section and so, should be removed from this section.

Ln270: 0.15?

Ln 274: It is already mentioned in the text that spectral binning was set to four for this study and so, the sentence can be discarded.

Ln289: Some of the information covered in this paragraph could be merged with the paragraph on Ln258 for a better flow and prevent repetition.

Ln302: Please revise HNO3

Ln308: It is not clear to me these grain sizes of the minerals were determined. Was an additional analysis performed on the soil samples before spectral analysis?

Figure 5-6-7: The captions indicate that maximum mineral percentages are given in the graphs while in Ln313 it was stated that the image percentages are calculated as the average of triplet of each soil type - please clarify.

Ln455: "whatever" is for colloquial use. Please consider revising. e.g. "Regardless of the cementing material,...

Discussions: While providing necessary information in explaining the observed changes in mineral distributions, overall this section is hard to follow. Also, this section can be given in a more succinct way. Please consider revising this section for better read and clarity.

In addition, although these processes can be helpful explaining the observed changes, it is not clear to me how this changes can be directly related to soil aggregates and their stability in particular.

Author Response

(The authors gave the same response as above.)

Round 2

Reviewer 1 Report

Dear Authors,

I advise you to do English check, especially in added parts of the manuscript. I also have some additional comments on your paper:

line 13 due to climate change?

line 15 – surface soil mineralogy

line 24 – soil surface mineralogy

line 72 – the reference to climate change is not clear. Do you refer to global climate change? If so, please provide the relevant reference or explanation. Because I do not see the link between  global climate change and short period moisture content variations.

Line 113  - I still believe that for reproducibility of your research you should add some details about how did you determine the amount of water for each treatment or provide a reference if the methodology is published.

line 373- decreased and increased ? please specify what increased or decreased clearer.

lone 558 – how the validation can be a drawback?

line 591 – surface mineralogy

Author Response

Dear Reviewer,

We want to thank you for your comments and the advice to do the English check.

Please find attached a summary of your comments and the revised manuscript.

Yours sincerely,

Irena Ymeti & co-authors
